# Sentiment Analysis on COVID-19-Related Social Distancing in Canada Using Twitter Data

**DOI:** 10.3390/ijerph18115993

**Published:** 2021-06-03

**Authors:** Carol Shofiya, Samina Abidi

**Affiliations:** Faculty of Computer Science, Dalhousie University, Halifax, NS B3H 1W5, Canada; abidi@cs.dal.ca

**Keywords:** COVID-19, Twitter, social distancing, sentimental analysis, SentiStrength, support vector machine, performance evaluation

## Abstract

Background: COVID-19 preventive measures have been an obstacle to millions of people around the world, influencing not only their normal day-to-day activities but also affecting their mental health. Social distancing is one such preventive measure. People express their opinions freely through social media platforms like Twitter, which can be shared among other users. The articulated texts from Twitter can be analyzed to find the sentiments of the public concerning social distancing. Objective: To understand and analyze public sentiments towards social distancing as articulated in Twitter textual data. Methods: Twitter data specific to Canada and texts comprising social distancing keywords were extrapolated, followed by utilizing the SentiStrength tool to extricate sentiment polarity of tweet texts. Thereafter, the support vector machine (SVM) algorithm was employed for sentiment classification. Evaluation of performance was measured with a confusion matrix, precision, recall, and F1 measure. Results: This study resulted in the extraction of a total of 629 tweet texts, of which, 40% of tweets exhibited neutral sentiments, followed by 35% of tweets showed negative sentiments and only 25% of tweets expressed positive sentiments towards social distancing. The SVM algorithm was applied by dissecting the dataset into 80% training and 20% testing data. Performance evaluation resulted in an accuracy of 71%. Upon using tweet texts with only positive and negative sentiment polarity, the accuracy increased to 81%. It was observed that reducing test data by 10% increased the accuracy to 87%. Conclusion: Results showed that an increase in training data increased the performance of the algorithm.

## 1. Introduction

The coronavirus outbreak that created mayhem, infecting millions of people across the globe, was declared a pandemic by the World Health Organization on 11 March 2020. Since there were no effective vaccines and no proper information resources available, the government and public health sectors took several non-pharmaceutical precautionary measures to contain the spread of infection and for contact tracing [1,2]. Some of these measures implemented across many countries include isolation, quarantine, emergency lockdown, wearing masks at public places, and social or physical distancing. Although these measures facilitate in reducing the spread of infection, they can cause a possible detrimental effect on public mental health [3,4,5]. A number of studies conducted worldwide have shown negative impact of COVID-19 on individuals’ psychological wellbeing [3,4,5].

Social distancing, also called physical distancing, is one of the most wide-spread public health precautionary measures taken in the face of pandemic. It is defined as maintaining a physical distance of approximately 6 feet from another person at public spaces to lower the spread of respiratory infections through respiratory droplets [6]. Unfortunately, many reports have shown people showing non-compliance to social distancing, which contributes to the spread of the virus [7,8]. Other studies have identified the differences between people who are compliant to those who are non-compliant to social distancing practices [9,10]. Understanding public sentiments about social distancing will help recognize unique problems faced by individuals when social distancing, especially with regards to their mental health. Such information can be crucial in developing appropriate and relevant information resources to help comply people with the social distancing requirement whilst minimizing its detrimental effect on their mental health [11]. 

Social media can be regarded as a useful source for understanding public lives and opinions on various aspects of the environments in which people live, especially in relations to numerous controversial issues that people routinely face [12]. Social platforms provide an opportunity for the public to express and share their feelings freely among others. Microblogging is a most popular tool utilized by millions across the globe to share their views with others [12]. Twitter is one such microblogging tool in which people express their views in the form of short texts of 140 characters, called microblogs. These microblogs along with other data about the users are collectively called tweet objects [13], and can be a rich source of data for researchers. This data can be processed, and various types of analysis can be performed to understand public opinion [12]. Sentiment analysis is an area of study that analyzes public opinions or sentiments which are expressed freely on social networking sites such as Twitter, Instagram, etc. [14]. In Canada, the total population was estimated to be approximately 37 million in 2020, of which more than 25 million use social networking sites [15,16]. Out of these social media users, 23% were active users of the Twitter application as of March 2020 [17]. 

We believe that performing sentiment analysis on Canadian tweets in relation to social distancing will provide a better perspective on Canadians sentiments regarding social distancing. In addition, analysis of these tweets may provide a different outlook on how these sentiments could influence the actions of others as well as assist the government in framing definitive actions at shaping public policies and procedures related to social distancing in combating COVID-19, whilst maintaining the mental wellbeing of individuals.

The main objective of our study is to analyze Twitter data using the SVM approach and to understand the sentiments of Canadians about any dominant or prevalent discourse around social distancing related to COVID-19.

## 2. Related Work

Sentiment analysis involves performing certain mathematical calculations to examine the sentiments of people towards a specific aspect or individual [18]. Subjectivity analysis, opinion mining, and sentiment classification are other related terms in the literature that although might have a specific meaning, nevertheless, are often used interchangeably with sentiment analysis [19,20].

Sentiment analysis can be performed by a lexicon-based approach like SentiStrength, Senti Word Net, linguistic inquiry word count (LIWC), affective norms for english words (ANEW); a machine-learning approach like naïve bayes (NB), multi-layer perceptron (MLP), multinomial naïve bayes (MNB), random forest (RF), Maximum Entropy, support vector machine (SVM) or a hybrid approach that uses both lexicon-based and machine learning approach [21]. Machine learning calculates sentiment polarity through statistical techniques that are highly dependent on the size of the dataset and is ineffective in handling negative and intensifying sentences as well as performs poorly in different domains. The lexicon-based approach, on the other hand, requires manual input of sentiment lexicons and performs well in any domain but fails to encompass complete informal lexicons. The hybrid approach will help to overcome the limitations of both approaches, thus enhancing performance, efficiency, and scalability [22,23]. Research has shown that using a hybrid approach not only accelerates accuracy and maintains stability but also provides better results than using one approach or one standard tool [23,24].

Jongeling et al. [25] used four tools—SentiStrength, NLTK, Alchemy, and Stanford NLP—to perform sentiment analysis. Using weighted kappa and adjusted rand index (ARI) as the measurement techniques, they showed that, although all tools are distant from manual labeling, SentiStrength ranked second after NLTK [25].

Ahmad et al. [21] analyzed the effectiveness of the SVM, using a popular data mining tool called WEKA, in identifying the sentiment polarity of text data. Performance of the output dataset was evaluated using pre-labeled datasets by measuring precision, recall, and F-measure, and then made comparisons using WEKA. The study revealed that a huge dataset is required for SVM and that the performance is highly dependent on the input data [21]. Han et al. [26] analyzed sentiments using latent semantic characteristics based on the probability for classification and applied SVM with fisher kernel to extemporize classification [26]. Naw Naw [12] conducted the sentiment analysis with SVM and K-NN classifiers. The training data was collected using Twitter API and pre-processed by transformation, negation handling, tokenizing, filtering, and normalization. Feature selection was done using term frequency-inverse document frequency (TF-IDF), which was the input feature for the classification model using SVM and K-NN classifiers. Tweets were classified as positive, negative, and neutral sentiments [12].

Rani and Singh [19] used SVM for sentiment analysis. Twitter data was collected using Twitter API. Data was preprocessed, features extracted using TF-IDF, and linear and kernel SVM was implemented to find the sentiments. F-score, precision, recall, and accuracy were used to measure performance. The results showed Linear SVM with higher accuracy over kernel SVM [19]. Godea et al. [27] studied the sentiments of people towards e-cigarettes using Twitter data. Apart from identifying sentiments, they also identified the diffusion of information related to e-cigs. They collected two months worth of tweets and manually annotated some tweets to gather opinion words. These words were checked for polarity using SentiStrength and combined with bag-of-words to extract features, which proved to be effective and better than SentiStrength [27].

As seen from the review of the literature, various approaches have been used to analyze sentiments through the application of the SVM algorithm. However, employing a hybrid approach by combining lexicon-based and machine learning was most successful in increasing accuracy, maintaining stability, thus providing better results. Therefore, we believe implementing hybrid approach would overcome the limits of employing lexicon or machine learning approach and improve performance, efficiency, and scalability of the model.

## 3. Materials and Methods

### 3.1. Data Collection

The Twitter data to conduct this study were extracted from an open-source freely available IEEE website [28]. This is because Twitter API does not allow access to stream old data for more than one week. The freely available dataset contained global tweets which were mostly geotagged and filtered using keywords related to COVID-19 such as “*corona*”, “*coronavirus*”, “*coronavirus*” until April 17, 2020. After April 18, 2020, additional filtering keywords including “*covid*”, “*#covid*”, “*covid19*”, “*covid19*”, “*covid-19*”, “*covid-19*”, “*sarscov2*”, “*sarscov2*”, “*sars cov2*”, “*sars cov 2*”, “*covid_19*”, “*covid_19*”, “*ncov*”, “*ncov2019*”, “*ncov2019*”, “*2019-**ncov*”, “*2019-ncov*”,“*#2019ncov*”, “*2019ncov*” were added to the tweet dataset [29]. This freely available tweet data contained only the tweet IDs of the users since the Twitter policy does not provide access to streaming complete tweets and publish to third parties. In addition, the tweet IDs were available only from 20 March 2020 [28].

The tweet IDs extracted from the IEEE website required hydration to capture complete tweet information called tweet objects. Each tweet object contains data such as tweet created time, tweet ID, tweet text, truncated, retweeted status, quoted status, location, user data, and more in JSON format [13]. The tweet IDs were hydrated using the DocNow hydrator tool, a Twitter hydrator tool built as a desktop application that allows hydration of tweets in JSON as well as CSV format [30]. This tool requires configuration using Twitter app keys that are available by creating a Twitter account and a Twitter app.

The hydrated tweets from 20 March 2020, to 20 April 2020, were downloaded into a CSV file. The tweets containing Canada in user location attribute and social distancing related keywords such as “social distancing”, “physical distancing”, “#social distancing”, “#physical distancing” in tweet texts were included in the final data. Those tweets that do not contain user location were excluded from the dataset. Extraction of these tweets was performed using the regular expression library available in python. The final data comprised only tweet texts from Canada containing social distancing keywords. The data collection process is illustrated in Figure 1.

Manual annotation of tweets was performed to explore and to better understand tweet texts, opinions, and identify the presence of additional social distancing keywords that could be used to extract more tweets.

### 3.2. Hybrid Approach for Analysis

This study performed sentimental analysis using a hybrid approach by employing the SentiStrength v2.3 tool, a lexicon-based approach, to analyze and retrieve sentiment polarity and then applying the SVM algorithm, a machine learning approach, for classification and analysis. The sentiment polarities extracted from SentiStrength were used as labels for training the SVM algorithm.

### 3.3. Sentiment Polarity Analysis

The SentiStrength tool was employed to detect sentiment polarity because the tool was built based on the lexicon-driven method and has the highest average human-level accuracy compared to other tools for Twitter analysis [31]. SentiStrength is a widely used standalone tool that operates independently without any other hardware or software [32]. Moreover, the tool detects emotions, emoticons, negating words, booster words, slang, and idioms. The SentiStrength tool provides 2 scores as an output for each text. One score represents positive emotion ranging from 1 to 5, where 1 indicates not positive and 5 indicates extremely positive. The other score represents negative emotion ranging from −1 to −5, where −1 indicates not negative and −5 indicates extremely negative [31]. As Thelwall et al. [31] are the creators of the SentiStrength tool, we employed their approach in classifying sentiment polarity into positive, negative, or neutral. According to them, a text is:

Positive if positive emotion score + negative emotion score > 0;

Negative if positive emotion score + negative emotion score < 0;

Neutral if positive emotion score =−negative emotion score and positive emotion score < 4 [25,31].

### 3.4. Data Cleaning and Feature Extraction

Tweet texts obtained after filtration and removal of duplicates were processed again to remove URLs, punctuations, Twitter handlers such as @user, and short words that are less than 3 characters. This data was then tokenized, lower-sized, and stemmed. After pre-processing texts, feature extraction was performed, which is necessary to filter out the irrelevant words that do not manifest any sentiments. Some of the most common weighted methods used to extract features are Bag-of-Words, TF-IDF, and Word2Vec [19]. TD-IDF weighs each word that occurs in the text and apply its respective TF and IDF score. TF measures how frequently a word occurs in text and IDF decreases the weight of terms that occur very frequently and increases the weight of terms that occur rarely. The product of these scores is called TD-IDF weight of a word. In general, the higher the weight the rarer the term in the given text and vice versa [33,34,35]. We used TF-IDF because it helps identify meaningful words that add value to text.

### 3.5. Support Vector Machine Classification

After pre-processing the data and feature extraction, the dataset was divided into training and test datasets containing sentiment polarities to train the dataset using SVM. SVM is a supervised machine learning algorithm [36]. Supervised machine learning is a robust algorithm that uses labels of observations during the training/learning process. These trained algorithms are then used to classify/predict unlabeled observations. Supervised machine learning contains both classification and regression algorithms. Classification algorithms identify the category of unlabeled observation, whereas regression predicts the value of a target variable based on the training algorithm. The goal of supervised machine learning is to detect links between attributes and the outcome variable [37]. SVM can perform classification, regression, as well as identification of outliers which are built on a statistical learning framework. SVM identifies the decision boundary that is far from the closest data points of all classes by converting the data and drawing a hyperplane using mathematical functions called kernels. There are 4 types of kernels. The linear kernel used for linear separation, radial basis function (RBF) kernel for non-linear separations with unknown data, sigmoid and polynomial kernels. Linear kernels are usually used when the number of features is more than data [36].

### 3.6. Evaluation of Performance

Confusion matrix, precision, recall, and F1 measure are the metrics used for evaluating the SVM algorithm [30,38,39].

## 4. Results

Of billions of tweets hydrated, only 795 tweets had social distancing keywords on tweet texts and Canada on user location. These tweets contained 166 duplicate texts because they were retweeted at different time frames which lead to the final 629 tweet texts. On analyzing sentiments of tweet texts using the SentiStrength tool and following Thelwall et al. [31] approach for sentiment classification, we found that 252 tweets revealed neutral sentiments, 220 tweets revealed negative sentiments and only 157 tweets demonstrated positive sentiments.

During the manual annotation of tweets for 10 days period, a total of 226 tweets, we observed that most tweets exhibited neutral expression and showed sentiments that could be categorized as anger, anxiousness, spreading alertness, humor, sharing facts, and spreading hope. Anxiousness and anger can be grouped as negative sentiments, while spreading alertness, humor, and sharing facts can be grouped as neutral sentiments, and spreading hope can be grouped as positive sentiments. Table 1 illuminates the manually labeled sentiment categories with examples.

The sentiment polarity obtained from the SentiStrength tool was used to train the SVM algorithm. The final dataset was divided into 80% training and 20% test data. On evaluating performance using RBF kernel, the accuracy obtained was 71%. When the performance of the algorithm was evaluated only with positive and negative sentiment polarity using the linear kernel, the accuracy raised to 81%.

Reducing the test data by 10% increased the accuracy to 87%. The results showed that an increase in training data with positive and negative sentiment polarity increased the accuracy. Table 2, Table 3 and Table 4 mention precision, recall, f1-score, and accuracy.

## 5. Discussion

Our study is directed towards analyzing and understanding the sentiments of Canadians towards social distancing related to COVID-19 using the SVM algorithm during the initial outbreak of the pandemic, where social distancing and other non-pharmaceutical precautionary measures were made mandatory for public health. To conduct this study, we concentrated on Twitter data, because the Twitter app provides easy access to data for researchers through API compared to Facebook [40], which is used by the majority of Canadians. A one-month period Twitter data was extracted from an open-source publicly available IEEE website. The extracted data contained tweet ID’s which were hydrated into CSV files. The hydrated data contained billions of tweets that were tailored specifically to Canada and with social distancing keywords. The final data obtained was only 629 tweets. From the final data, 10 days of tweets, a total of 226 tweets, were manually annotated and divided into categories based solely on the author’s viewpoint. Each tweet text was assessed to identify words that could indicate spreading hope in texts like “encourage”, “Thank you”, “practice”; words that could indicate anxiousness or anger in texts like “pain”, “fail”, “wretched”; and words spreading alertness, humor and sharing facts in texts like “prepare”, “maintain”, “fact” and more. These could be further divided into positive, negative, or neutral based on orientation of the expressed sentiment in terms of its polarity. These manually annotated tweets were categorized just for our own understanding and have not been employed during analysis. The manual exploration of tweets helped us to understand more about public concerns related to the pandemic, diffusion of available COVID-19 resources, creating awareness about the preventive measures, staying positive, and tweeting funny stories about ongoing events.

The SentiStrength tool was exploited to find the sentiment polarity of tweets which was later used as labels to train the SVM algorithm. The dataset was divided into 80% training and 20% testing data. When the entire dataset with positive, negative, and neutral sentiments were used, the accuracy obtained using the RBF kernel of the SVM algorithm was 71%. The accuracy did not improve upon using the RBF kernel for only positive and negative sentiments. However, the accuracy increased to 81% upon using linear kernel for positive and negative sentiments. We found that increasing the training data with only positive and negative sentiments increased the accuracy to 87%.

This study provided better insights into the Canadians perspective towards social distancing which can be implemented to make better public health decisions and frame government policies. Twitter can act as a knowledge translation tool disseminating knowledge to a large group of people. Along with broadcasting COVID-19 information resources, public health can also consider spreading a sense of optimism among people through positive motivating videos, pictures, games, and more. In addition, public health can continue monitoring people with negative sentiments and deliver targeted information to preserve the mental health of those individuals who show extremely negative sentiments. Further, the number of positive and negative sentiments can help public policymakers assess compliance and non-compliance in practicing preventive measures. This can assist policymakers to make necessary changes that promote people to follow preventive measures.

The output obtained from the SentiStrength tool, a traditional classification method, was almost similar to the observation made during manual annotation. Most Twitter users were found neutral towards social distancing. Manual annotation of data and the use of the SentiStrength tool along with the application of the SVM algorithm was distinctive and we were able to better understand the data.

The Twitter data obtained from the dataset had numerous variables like the date and time, retweets, user profile information, locations, and more. A lot of predictive and geospatial analytics can be performed using this data. This approach can also be used to extract a large number of tweets after 20 April 2020 and apply other classifiers to perform other analysis such as predictive and geospatial analysis.

## 6. Limitations

This study has several limitations. Firstly, the size of the dataset directly influenced the performance of the SVM algorithm. An increase in training data would have increased the accuracy. Secondly, hydration of tweet IDs into CSV files resulted in the deletion of approximately 20% of tweets. This resulted in the loss of some data which could have influenced tweet data extraction and thus influencing the results. The dataset also had missing tweets from 29 March 2020, 04:02 PM to 30 March 2020, 02:00 PM due to some technical faults. Thirdly, almost 40% of tweets showed neutral sentiments that could be one of the reasons for reduced accuracy. Fourthly, the performance of the SentiStrength tool was not evaluated as the polarity extracted from this tool had been used as labels to train the SVM algorithm. Finally, the dataset might have opinion spamming and dual opinion tweets.

## 7. Conclusions

This study analyzed the sentiments of Canadians towards social distancing related to COVID-19 for one month using Twitter data. SentiStrength tool and SVM Classifier were exploited to perform the analysis. The result showed that 40% of Canadians showed neutral sentiments towards social distancing followed by 35% showed negative sentiments and only a quarter of Canadians were positive towards social distancing. Performance evaluation of the SVM algorithm resulted in 87% of accuracy, which could be increased by increasing the training data. A large dataset is required to increase the performance of the algorithm.

## Figures and Tables

**Figure 1 ijerph-18-05993-f001:**
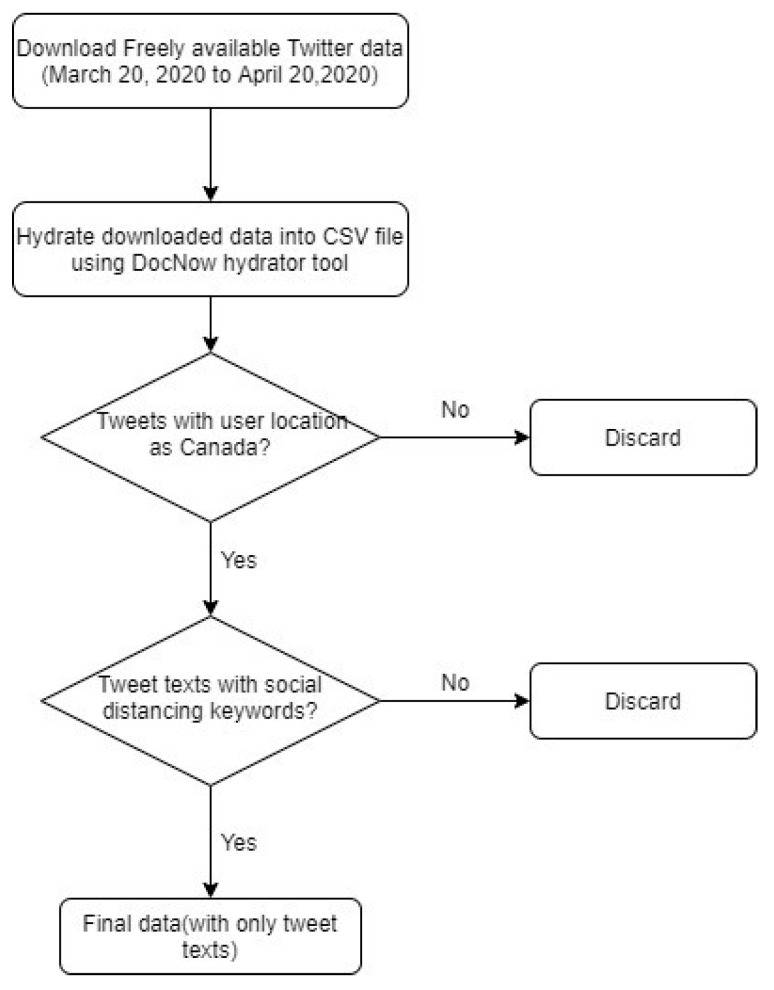
Data collection process.

**Table 1 ijerph-18-05993-t001:** Manually labelled sentiment categories with examples.

Example	
Anxiousness	Is it safe to cycle? Why social distancing is important, and gathering needs to be avoided as prevention against Corona &amp. Looking for someone going out amidst Corona Virus & social distancing to help people who are experiencing https://t.co/Ll6wcWRgtR homeless, accessed on 11 April 2021, elderly people, people laid off work/lost their jobs. Uplifting experiences during this difficult time. Corona is droplet not airborne itâ€^TM^s why social distancing is being stressed The Coronavirus outbreak and social distancing has been a challenge for everyone. In lots of cities, weâ€^TM^re seeing an uptickâ€¦
Anger	So OAU has not adhered to social distancing? When will you people start taking corona-virus serious? Government acting contrary to its directives. Is there social distancing here ðŸ‘‡ðŸ‘‡. So are we preventing spread of corona virus or we creating a breeding ground? I hate reading this. I found out my Aunt has the Corona virus last week, and she said she was coughing so badly she couldn’t sleep. The one piece of advice she gave me was to take social distancing seriously. So yeah. Stop going out!
Spreading alertness	Let us practice social distancing to fight and stop corona pandemic This is EXTREMELY important. Stay home, continue social distancing. DO NOT GET COMPLACENT and DO NOT assume this isn’tâ€¦ #staysafeoutthere the #corona is real af #welovehiphopðŸŽ^TM^ #worldsmostsmokedoutpod-castðŸŒðŸ’¨#bluntstothehead #socialdistancing #movethecultureforwardâ€ 1 ï¸https://t.co/EOOqZc1dMh
Humor	Most responsible drinkers on the earth are from Kerala, while maintaining a social distancing in the view of Corona, lookâ€¦ I’m watching Tangled and I can’t believe Rapunzel practiced social distancing in a tower away from the village of Corona.â€¦ This wife has had enough of her husband and this quarantine! ðŸ~©ðŸ~,ðŸ~, #Socialites, is your partner driving you nuts too?ðŸ‘‡ðŸ 3 ðŸ“¸: onefunnymommy #likeforlikes #corona #quarantine #socialdistancing #covid_19 #coronavirus #coronaâ€¦ Due to Corona virus and social distancing, Singer Kelis has announced that her milkshake will now bring no more than ten boysâ€¦
Sharing facts	Very good article on social distancing: Why outbreaks like coronavirus spread exponentially, and how to â€œflatten the curveâ€ |â€¦ Weâ€^TM^ve seen many explanations of the exponential spread of COVID-19 and how social distancing can flatten it. This dynamic simâ€¦ I strongly encourage folks to read and view this (in Chrome).
Spreading hope	Let us practice social distancing to fight and stop corona pandemic Thank you residents for your help in reducing crowding and exercising â€œsocial distancingâ€¦ Likely the best simple educational short vid to date on the corona virus. This is worth sharing to motivate social distancing and clear up misinformation.

**Table 2 ijerph-18-05993-t002:** Performance evaluation of final data with positive, negative, and neutral sentiments.

Sentiment Polarity	Precision	Recall	F1-Score	Support
Negative	1.00	0.25	0.40	20
Neutral	0.67	1.00	0.81	62
Positive Accuracy = 71%	1.00	0.32	0.48	22

**Table 3 ijerph-18-05993-t003:** Performance evaluation of data set with positive, negative sentiments with 20% test data.

Sentiment Polarity	Precision	Recall	F-score	Support
Negative	0.73	1.00	0.85	41
Positive Accuracy = 81%	1.00	0.62	0.77	40

**Table 4 ijerph-18-05993-t004:** Performance evaluation of data set with positive, negative sentiments with 10% test data.

Sentiment Polarity	Precision	Recall	F1-Score	Support
Negative	0.85	0.96	0.90	24
Positive Accuracy = 87%	0.93	0.76	0.84	17

## Data Availability

All data are fully available. Open-source publicly available covid-19 geo-tagged tweets data by Rabindra Lamsal from the IEEE website was accessed.

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
