# Peer review of "Sentiment Analysis on COVID-19-Related Social Distancing in Canada Using Twitter Data"

_ijerph, 2021, doi:10.3390/ijerph18115993_

Round 1

Reviewer 1 Report

This is an interesting paper, one that can add to the discourse on COVID-19 regarding social distancing. As a reviewer, my main concern is the presentation of the content. Significant and comprehensive editing of the paper throughout is required. I have a few other comments listed by paragraph as follows:

S1, third paragraph: The research may provide information that can inform policy and practices, but it is limited by the focus on Twitter. Do people using this microblog represent the broader Canadian population?

S3, 2nd paragraph: Please expand on the use of hybrid tools. What is it exactly that makes these better?

S4.1, 3rd paragraph: The third sentence looks out of place, please reword.

S4.3, 1st paragraph: The the1wall discussions requires clarification and elaboration for reader comprehension.

S4.5, Please expand on the meaning of supervised machine for reader clarification.

S5, 2nd paragraph: provide references for the use of sentimental categories (unless this is predetermined by the technology).

S6. The discussion is very thin and can be expanded to highlight how the research can inform policy and practice. Specific examples are needed here if the research is to have the impact that you claim at the outset.

As indicated above the article requires comprehensive editing -- grammar, syntax, use of terms and so forth so that readers can fully comprehend the research.

Thank you for the interesting work. I wish you well with the revisions.

Author Response

Point 1: S1, third paragraph: The research may provide information that can inform policy and practices, but it is limited by the focus on Twitter. Do people using this microblog represent the broader Canadian population?

Response added to the manuscript: under section 1(Introduction), third paragraph, 8th and 9th sentence as well as under section 5 (Discussion), first paragraph, 2nd sentence.

Response 1: The total Canadian population was estimated to be approximately 37 million in 2020, of which more than 25 million use social networking sites. In March 2020, 70% of social network users were using Facebook, ranking the highest, while only 23% were using Twitter. Although most Canadians use Facebook, the Twitter app provides easier access to researchers to download data containing complete user information through API compared to Facebook that has privacy concerns. Hence, we concentrated on extracting text data from Twitter for our analysis.

Point 2: S3, 2nd paragraph: Please expand on the use of hybrid tools. What is it exactly that makes these better?

Response added to the manuscript: under section 2(Related work), second paragraph, 2nd,3rd,4th sentence as well as sixth paragraph, 2nd and 3rd sentence.

Response 2: Hybrid approach combines machine learning, enhancing accuracy and lexicon-based approach, increasing efficiency and scalability. Machine learning calculates sentiment polarity through statistical techniques which are highly dependable on the size of the dataset and is not effective in handling negative and intensifying words as well as displays poor performance when domain differs. The lexicon-based approach, on the other hand, requires manual input of sentiment lexicons and performs well on any domain but fails to encompass all informal sentiment lexicons. Using a hybrid approach will help to overcome the limitations of both approaches enhancing performance, efficiency, and scalability.

Point 3: S4.1, 3rd paragraph: The third sentence looks out of place, please reword.

Response added to the manuscript: under section 3.1(Data collection), third paragraph, 3rd sentence.

Response 3: Those tweets that do not contain user location were excluded from the dataset.

Point 4: S4.3, 1st paragraph: The the1wall discussion requires clarification and elaboration for reader comprehension.

Response added to the manuscript: under section 3.3(sentiment polarity analysis), 1st paragraph, 4th, 5th, 6th, 7th, 8th sentence.

Response 4: SentiStrength tool provides 2 scores as an output for each text. One score represents positive emotion ranging from 1 to 5 where 1 indicates not positive and 5 indicates extremely positive. The other score represents negative emotion ranging from -1 to -5 where -1 indicates not negative and -5 indicates extremely negative. As Thelwall et.al were the creators of the SentiStrength tool, we employed their approach to mapping sentiment polarity into positive, negative, and neutral. According to them, a text is:

Positive if positive + negative >0

Negative if positive + negative <0

Neutral if positive = -negative and positive < 4

Point 5: S4.5, Please expand on the meaning of supervised machine for reader clarification.

Response added to the manuscript: under section 3.5(Support Vector Machine Classification), first paragraph, 3rd, 4th, 5th, 6th, 7th sentence.

Response 5: Supervised machine learning is a robust algorithm that uses labels of observations during the training/ learning process. These trained algorithms are then used to classify/ predict unlabeled observations. Supervised machine learning contains both classification and regression algorithms. Classification algorithms identify the category of unlabeled observation based on training algorithm whereas regression algorithms predict the value of a target variable based on the training algorithm. The goal of supervised machine learning is to detect links between attributes and the outcome variable.

Point 6: S5, 2nd paragraph: provide references for the use of sentimental categories (unless this is predetermined by the technology).

Response added to the manuscript: under section 5(Discussion), 1st paragraph, 7th, 8th, 9th, 10th, 11th sentence.

Response 6: The sentiment categories were made based on manual annotation performed by authors and grouped accordingly and therefore we are unable to provide references.

Point 7: S6. The discussion is very thin and can be expanded to highlight how the research can inform policy and practice. Specific examples are needed here if the research is to have the impact that you claim at the outset.

Response expanded on the manuscript: section 5(Discussion), 3rd paragraph, 2nd, 3rd, 4th, 5th, 6th sentence.

Reviewer 2 Report

The authors conducted a sentiment analysis on tweets from Canada regarding social distancing. Sentiment was evaluated using SentiStrength and sentiment classification was performed using support vector machine. The idea is interesting but in my opinion the manuscript would require several changes to improve its scientific soundness. 

While section 3 on related work reports some literature on different lexicon-based and machine learning methods that can be used to conduct a sentiment analysis, and while the authors suggest there might be an added value in using both methods, it is not clear to me why specifically SentiStrength and SVM were used. Especially in the case of SentiStrenght, the rationale of the choice should be highlighted.  

I don't understand how the lexicon-based and SVM method were integrated. Did the authors used sentiment scores provided by SentiStrenght as labels to train the SVM? Or did they use manually-labeled tweets? This information is missing from the methods and should be integrated. 

I understand that the source used by the authors made tweets available only from 20 March 2020. However, whey did the authors choose to only analyze tweets to 20 April 2020? This should be explained.

The authors should include a list of "social distancing" keywords used to select tweets to include.

Why was manual annotation of specific performed only on a subset of tweets (10 days?). Also, more details on this should be provided (e.g. how many tweets were manually annotated and by how many examiners). 

I think the manuscript is missing a section in which the two methods to evaluate sentiment are compared. We know the accuracy of the SVM method but no evaluation of the SentiStrenght tool. In which sense using both methods improved the results?

The discussion should be largely expanded as at present it doesn't really discuss the results. 

As a final note, I would change "sentimental analysis" in "sentiment analysis" in the title and English language should be revised throughout the text as several grammar errors are present. 

Author Response

Response to Reviewer 2 Comments

Point 1: While section 3 on related work reports some literature on different lexicon-based and machine learning methods that can be used to conduct sentiment analysis, and while the authors suggest there might be an added value in using both methods, it is not clear to me why specifically SentiStrength and SVM were used. Especially in the case of SentiStrength, the rationale of the choice should be highlighted.  

Response added to the manuscript: under section 3.3(Sentiment  polarity analysis), 1st paragraph, 1st, 2nd and 3rd sentence.

Response 1: SentiStrength tool was employed because the tool was built based on a lexicon-based approach and has the highest average human-level accuracy compared to other tools for Twitter analysis. This is a widely used stand-alone application that can operate independently of hardware or software. Moreover, the tool can detect emotions, emoticons, negating words, booster words, slang, and idioms.

Point 2: I don't understand how the lexicon-based and SVM methods were integrated. Did the authors use sentiment scores provided by SentiStrength as labels to train the SVM? Or did they use manually labeled tweets? This information is missing from the methods and should be integrated. 

Response added to the manuscript: under section 3.2(Hybrid  approach for analysis), 2nd sentence and section 4(Results), 3rd paragraph, 1st sentence as well as section 5(Discussion), 1st paragraph, 10th, 11th sentence and 2nd paragraph, 1st sentence.

Response 2: The authors used the SentiStrength tool, a lexicon-based approach to extract sentiment polarity of tweets, and these sentiment polarities were used as labels for training the SVM algorithm.

Point 3: I understand that the source used by the authors made tweets available only from 20 March 2020. However, why did the authors choose to only analyze tweets to 20 April 2020? This should be explained.

Response added to the manuscript: under section 5(Discussion), 1st paragraph, 1st sentence and 5th paragraph, 3rd sentence.

Response 3: Our focus was to assess sentiments of Canadians towards social distancing during the initial period of Covid-19 as social distancing is not a usual practice during normal day-to-day activities. Analyzing tweets after 20 April 2020 is beyond the scope of this study. However, future studies can include extracting tweets after 20 April 2020.

Point 4: The authors should include a list of "social distancing" keywords used to select tweets to include.

Response added to the manuscript: under section 3.1(Data collection), 3rd paragraph, 2nd sentence.

Response 4: The social distancing keywords used to extrapolate tweets are “social distancing”,” physical distancing”, “#socialdistancing”, “#physicaldistancing”.

Point 5: Why was a manual annotation of specific performed only on a subset of tweets (10 days?). Also, more details on this should be provided (e.g., how many tweets were manually annotated and by how many examiners). 

Response added to the manuscript: under section 4(Results), 2nd paragraph, 1st sentence and under section 5(Discussion), 1st paragraph, 6th sentence.

Response 5: Both examiners manually annotated 10 days tweets, a total of 226 tweets. We assumed that manual annotation of 10 days tweets would suffice manual exploration of tweet texts to get a basic understanding of the opinion of Canadians towards social distancing.

Point 6: I think the manuscript is missing a section in which the two methods to evaluate sentiment are compared. We know the accuracy of the SVM method but no evaluation of the SentiStrength tool. In which sense using both methods improved the results?

Response added to the manuscript: under section 6(Limitations), 8th sentence.

Response 6: Since we used the SentiStrength tool to identify sentiment polarity and these labels were used to train the SVM algorithm. We specifically did not evaluate the SentiStrength tool and is one of the limitations of this research study.

Point 7: The discussion should be largely expanded as at present it doesn't really discuss the results.

Response expanded on the manuscript: under section 5(Discussion), 1st and 2nd paragraph

Round 2

Reviewer 1 Report

Thank you for the edits, I wish you well with the publication.

Reviewer 2 Report

The authors addressed all questions. 

In the 3.4 section, please replace "Tweet texts obtained after filtration and removal of duplicates was processed ..." with "Tweet texts obtained after filtration and removal of duplicates were processed ..."
